# Eosinophilic Cholecystitis and Eosinophils in Gallbladder Injuries: A Clinicopathological Analysis of 1050 Cholecystectomies

**DOI:** 10.3390/diagnostics13152559

**Published:** 2023-08-01

**Authors:** Bahar Memis, Burcu Saka, Juan Carlos Roa, Sudeshna Bandyopadhyay, Michelle Reid, Pelin Bagci, Berk Kaan Aktas, Ayse Armutlu, Olca Basturk, N. Volkan Adsay

**Affiliations:** 1Department of Pathology, Sisli Hamidiye Etfal Training and Research Hospital, Istanbul 34396, Turkey; 2Department of Pathology, Koc University, Istanbul 34450, Turkey; 3Department of Pathology, Pontificia Universidad Catolica de Chile, Center for Cancer Prevention and Control (CECAN), Millennium Institute on Immunology and Immunotherapy (IMII), Santiago 8331150, Chile; 4Department of Pathology, Wayne State University, Detroit, MI 48202, USA; 5Department of Pathology, Emory University, Atlanta, GA 30322, USA; 6Department of Pathology, Marmara University, Istanbul 34854, Turkey; 7Department of Pathology and Laboratory Medicine, Memorial Sloan Kettering Cancer Center, New York, NY 10065, USA

**Keywords:** cholecystectomy, cholecystitis, acute, subacute, chronic, eosinophilic, eosinophils, histopathology

## Abstract

“Eosinophilic cholecystitis” has been an elusive concept. Around 1050 consecutive cholecystectomies with chronic (CC, *n* = 895), subacute (SAC, *n* = 100), and acute cholecystitis (AC, *n* = 55) were reviewed for eosinophilic infiltration. Eosinophilic hot spots (>40 eosinophils/HPF) were seen in 63% of SAC and 35% of AC (vs. 6% of CC, *p* < 0.001). Eosinophils were mostly encountered in areas of wall thickening, revealing edema with early collagenization and young tissue-culture-type fibroblasts. However, in ten chronic cholecystitis patients (<1%), prominent eosinophilia with eosinophil-rich foci (>100 eosinophils/HPF) was noted. These ten cases, classified as “eosinophilic cholecystitis”, were analyzed further: The patients were relatively young (mean age = 43 years), with a 9:1 female:male ratio. None had blood eosinophilia/eosinophilia syndromes. Although one had ulcerative colitis, others did not have any autoimmune diseases. The mean gallbladder wall thickness was 3.5 mm (vs. 4.2 mm in ordinary CC). In conclusion, eosinophils are a part of especially subacute injuries in the gallbladder. They are typically condensed in the areas of healing and appear to signify a distinctive state of injury in which there are erosions leading to slow/sustained exposure of the mural tissues to the bile contents that induce chemical injury/recruit eosinophils. Eosinophilic cholecystitis is a very uncommon occurrence and appears to be an exaggerated response in allergic patients who are prone to recruit eosinophils in reaction to injury.

## 1. Introduction

An entity designated “eosinophilic cholecystitis” is recognized as a distinct type of cholecystitis [1,2]. Since its first description in 1949 [3], it has been reported in a limited number of case series [4,5,6,7,8,9,10], with an incidence ranging from 0.02 to 6.4% of cholecystectomies [4,5,7,9]. Being clinically indistinguishable from the other types of cholecystitis, the diagnosis of eosinophilic cholecystitis is primarily histological and performed via analysis of the cholecystectomy specimen [4]. However, the term eosinophilic cholecystitis has been used variably, and there has been no uniform definition for the pattern of injury or characterization of its clinicopathologic associations. Some investigators have defined eosinophilic cholecystitis based on the density of eosinophils [5,11], while others have also considered which gallbladder layers are involved as part of the diagnosis [6,7,12,13]. More importantly, although eosinophils are known as participants in some phases of acute cholecystitis [1,2,14,15], most of the case reports and series have included subacute cases as eosinophilic cholecystitis as well [5,6,7,10], and some authors have employed the term in the setting of chronic cholecystitis [16]. 

In this retrospective study, a series of 1050 consecutive cholecystitis specimens were reviewed to determine the presence, amount, pattern, and clinicopathological associations of eosinophilic infiltration in various types and phases of gallbladder injury.

## 2. Materials and Methods

This study was conducted in full accordance with local GCP guidelines and current legislation, while permission was obtained from the institutional ethics committee (Institutional Review Board of Emory University) for the use of patient data for publication purposes (IRB name—Clinicopathologic and Molecular Analysis of Gallbladder and Biliary Tract Lesions; IRB approval number—IRB00010713; date of approval—23 May 2017).

### 2.1. Specimens

A total of 2337 consecutive, routinely sampled cholecystectomy specimens related to non-obstructive pathologies (*n* = 1260; 1050 cholecystitis; 210 non-neoplastic polyps), obstructive pathologies in the distal pancreatobiliary tract [*n* = 128; 62 distal common bile duct/pancreatic tumors, 35 primary sclerosing cholangitis, and 31 autoimmune pancreatitis], and gallbladder neoplasia, including dysplasia, intracholecystic papillary neoplasms or invasive carcinomas (*n* = 949) were identified at Emory University Hospital between 2002 and 2010.

Original H&E-stained histologic sections (an average of 2 slides per case) of 1050 cholecystitis cases accessioned were retrieved and reevaluated to determine the patterns of injury as acute, subacute, and chronic. They were also evaluated for the presence, amount, and distribution of eosinophilic infiltration.

Cholecystectomies with non-neoplastic polyps or neoplastic lesions such as dysplasia, intracholecystic papillary neoplasms, invasive carcinomas, or cholecystectomies performed during pancreatoduodenectomies were excluded. 

### 2.2. Patterns of Injury

Acute cholecystitis was defined as a gallbladder with overt edema, congestion/hemorrhage, effacement of the normal histologic elements, mucosal/transmural necrosis, vasculitis, or ulcer with/without an acute inflammatory reaction [1,2,17].

Subacute cholecystitis was defined as a gallbladder revealing more subtle foci of young tissue culture-type fibroblasts in a setting of edema and thickening of the wall with new capillarization. Reactive changes in the serosal layer were also a common feature [1,2,17].

Chronic cholecystitis was defined as a gallbladder that was characterized by thickening of the GB wall with fibrosis and/or muscular hypertrophy (but not young tissue culture-type fibroblasts), with/without gallstones, with/without chronic inflammation, and with/without reactive, metaplastic changes in the epithelium [1,2,17].

### 2.3. Assessment of Eosinophils

A standard Olympus BX41 microscope was used to identify eosinophils in each case. All cases were screened at 100× magnification for the presence of eosinophils; areas with eosinophils were examined at 400× magnification and graded as negligible (<40 eosinophils per HPF) or noticeable (>40 eosinophils per HPF). Cases with intense eosinophilia (>100 eosinophils per HPF in multiple fields) were qualified as “eosinophilic cholecystitis”. Assessment of eosinophils took, on average, 30 min per case.

### 2.4. Correlative Parameters

Patients’ demographic data (age and gender), gallstone information, blood eosinophil levels, clinical history of parasitic infestation, asthma, any other allergic or autoimmune diseases, or potentially related chronic diseases (such as eosinophilic gastroenteritis, eosinophilia-myalgia syndrome, tryptophan-induced eosinophilia, eosinophilic granulomatosis with polyangiitis, and inflammatory bowel diseases) were recorded from patients’ charts. 

Gallbladder wall thickness was recorded from the original pathology reports and correlated with microscopic measurements. 

Assessment of correlative parameters took, on average, 90 min per case.

### 2.5. Statistical Analysis

Statistical analysis was made using IBM SPSS Statistics for Windows, version 22.0 (IBM Corp., Armonk, NY, USA). One-way analysis of variance (ANOVA) and Post-Hoc Games-Howell tests were used for the analysis of numerical data. Chi-square test was used to analyze categorical data. Data were expressed as “mean” and “percent (%)” where appropriate. *p* < 0.05 was considered statistically significant.

## 3. Results

### 3.1. Incidence

Of the 1050 consecutive cholecystectomies, 895 cases were classified as chronic cholecystitis, 100 as subacute cholecystitis, and 55 as acute cholecystitis. 

Noticeable eosinophil infiltration (hot spots with >40 eosinophils per HPF) was significantly more frequent in subacute cholecystitis and acute cholecystitis groups than in chronic cholecystitis groups [63 of 100 cases (63%) and 19 of 55 cases (35%) vs. 52 of 895 cases (6%), respectively, *p* < 0.001].

However, massive eosinophilic infiltration that was qualified as “eosinophilic cholecystitis” (>100 eosinophils per HPF) was only seen in the chronic cholecystitis group [10 of 895 cases, (1.1%)]. Overall, eosinophilic cholecystitis was identified in 0.95% of all (1050 cases) cholecystectomies.

### 3.2. Clinical Associations 

Eosinophilic cholecystitis patients were relatively younger (mean age 43 years vs. 49 years in ordinary chronic cholecystitis patients) and almost exclusively female (F/M of 9 vs. 3 in ordinary chronic cholecystitis) (Table 1). 

Eosinophilic cholecystitis was associated with a history of allergies; four of six cases with a patchy pattern and two of four cases with a diffuse pattern (see below) had a history of asthma and drug reactions. Another patient had a positive stool test for Giardiasis. None of the patients had elevated blood eosinophil levels. Although one patient had ulcerative colitis, others did not have a history of an autoimmune or potentially related chronic disease (such as eosinophilic gastroenteritis, eosinophilia-myalgia syndrome, tryptophan-induced eosinophilia, or eosinophilic granulomatosis with polyangiitis).

### 3.3. Pathologic Findings

In acute cholecystitis, noticeable eosinophil infiltration was identified in the hemorrhagic foci (Figure 1A). In 13 cases (25%), secondary vasculitis characterized by eosinophils in the endothelium and subendothelium was also observed, particularly in the vasculature of perimuscular connective tissue (Figure 1B). No necrotizing vasculitis was present.

In subacute cholecystitis, noticeable eosinophil infiltration was concentrated in areas of healing/healed erosions, showing denuded epithelium (Figure 2A), in areas of impacted stone (Figure 2B), and in foci of subacute changes, which were characterized by edema and prominent young tissue culture-like fibroblasts (Figure 2C).

In chronic cholecystitis, while eosinophils were noted in 125 cases (14%), they were in the form of very sparse few-cell clusters, usually localized to the deep edges of the tunica muscularis (Figure 3) in 73 cases (8%). Scattered perivascular cuffs around the vessels without the involvement of the endothelium or subendothelium were also present. Noticeable eosinophil infiltration was detected only in 52 cases (6%).

Of the ten cases that qualified as eosinophilic cholecystitis, six revealed multiple scattered foci of intense eosinophilia (>100 eosinophils per HPF) (patchy pattern) (Figure 4). All (100%) were associated with gallstones; however, they had relatively uninjured walls and the mean gallbladder wall thickness was 3.5 mm (vs. 4.2 mm in ordinary chronic cholecystitis) (Table 1). Intraepithelial eosinophils were detected in only one case. 

The remaining four cases had sheets of intense eosinophilia (>100 eosinophils per HPF) to the exclusion of other inflammatory cells (diffuse pattern) (Figure 5). Their gallstone rate was low (50%) compared to all other groups, and they had a relatively thin gallbladder wall with a mean wall thickness of 3.3 mm (vs. 4.2 mm in ordinary chronic cholecystitis). Intraepithelial eosinophils were not detected, but the distribution was otherwise transmural.

Clinicopathologic features of the groups are summarized in Table 1.

## 4. Discussion

In the gallbladder, eosinophils are encountered predominantly in subacute cholecystitis. Analysis of eosinophils in three patterns of gallbladder injury in our study indicates that eosinophils do not seem to have a significant role in chronic cholecystitis, as noticeable eosinophil infiltration is far less common in these cases (6%) compared to subacute (63%) and acute (35%) gallbladder injury (*p* < 0.001). Moreover, the affinity of eosinophils to concentrate in the areas with healing erosions (but without overt destruction) also suggests that breached epithelial integrity may lead to slow and sustained exposure of the mural tissues to the chemical effects of the bile and the subsequent chemotaxis of eosinophils. Therefore, in the right setting, eosinophils can be considered a sign of a subacute process. A similar pathogenesis is also suggested for drug-related (e.g., erythromycin and ampicillin) eosinophilic cholecystitis, characterized by intraepithelial eosinophils and evidence of response to excreted metabolites (i.e., antigen) [11].

Massive (>100 eosinophils per HPF) infiltration of the gallbladder wall by eosinophils, classified here as “eosinophilic cholecystitis”, on the other hand, is a very rare occurrence in this organ. In this study, this phenomenon was observed only in the context of chronic cholecystitis, and its frequency was <1% of all cases, which is in accordance with some of the other reports in the literature [7,9], although some studies have kept a lower threshold and reported the incidence much higher [5,6].

Our study also establishes that eosinophilic cholecystitis occurs almost exclusively in women, something that was noted in some other studies in the literature [6,7], but not all [9]. Moreover, eosinophilic cholecystitis appears to be a condition of relatively younger women, as it is seen mainly between ages 35 and 45, and this was also observed in some studies in the literature [5,7,11,14,18,19,20]. In fact, some studies have even reported patients who were teenagers or young adults [6,10]. However, this condition is seldom reported in advanced age [9,21]. This suggests that a competent and perhaps hyperactive immune system may be required to mount this kind of response. 

Although eosinophilic cholecystitis was first defined in two acalculous patients [3] and has been reported to manifest more frequently in acalculous cholecystitis than in cholelithiasis in some studies [5,22,23], even up to three times [4], concomitant gallstone reports are also present. A retrospective analysis of 7494 cholecystectomy specimens revealed that 12 patients (0.2%) were diagnosed with eosinophilic cholecystitis, and all of them had gallstones [7]. In another retrospective study of 1370 cholecystectomy specimens, eosinophilic cholecystitis was diagnosed in 22 (1.6%) patients, and gallstones were present in most patients [6]. In our cohort, this condition often occurred in association with gallstones; however, half of the cases with more diffuse gallbladder involvement lacked gallstones. Therefore, it is possible that in some cases, an inherent predisposition rather than a process triggered by cholelithiasis may be at play. 

While the etiology of eosinophilic cholecystitis remains unknown, associations with allergies without any other concomitant medical conditions [6,14], as well as use of herbal medications, hypersensitivity to drugs, parasitic infections, autoimmune diseases, hypereosinophilic syndrome with concomitant eosinophilic infiltration of the other sites of the gastrointestinal tract, and eosinophilic granulomatosis with polyangiitis, have been reported [24,25,26,27,28,29]. In our series, a history of allergies is noted in six of ten eosinophilic cholecystitis cases. Also, one case has giardiasis, and another has ulcerative colitis. Of note, blood eosinophilia has been reported in 9% to 26% of patients with eosinophilic cholecystitis, likely as a manifestation of parasitic infections or hypereosinophilic syndrome [13,24,25,26,27,28,29,30,31,32]. However, acute cholecystitis is also considered one of the reasons for hypereosinophilia [33], and some authors define eosinophilic cholecystitis as the resolution phase of acute cholecystitis with conspicuous eosinophils. Therefore, the relationship between eosinophilic cholecystitis and blood eosinophilia has been debatable. None of our patients have peripheral eosinophilia.

More importantly, since eosinophilic cholecystitis has similar clinical and laboratory manifestations to ordinary chronic cholecystitis, it is still being questioned whether eosinophilic cholecystitis needs to be recognized as a separate entity. There are reports stating that eosinophilic cholecystitis patients should probably not be considered as suffering from a specific form of gallbladder disease but as having cholecystitis with an unusual cellular response [16]. In contrast, other authors recommend a complete work-up of the patient to exclude other organ involvements and associated syndromes [6,8]. 

Interestingly, eosinophilic cholecystitis cases in our series have relatively fewer injured gallbladders with no significant wall thickening compared to ordinary cholecystitis (3.5 mm vs. 4.2 mm). This is compatible with the literature indicating the absence of fibrotic and destructive changes as a notable distinguishing feature of eosinophilic cholecystitis [16]. This finding seems intriguing given the association of eosinophils and fibrotic conditions of different etiopathology, including scleroderma, endomyocardial fibrosis, idiopathic pulmonary fibrosis, asbestos-induced lung fibrosis, wound repair, and tissue remodeling, as well as eosinophilic cholangitis [34]. 

Also interesting, but perhaps not surprisingly, a history of allergic conditions is common in our patients with eosinophilic cholecystitis. On the other hand, our patients also have gallstones, and therefore it is difficult to determine whether the eosinophils are primary instigators of injury or whether they are in abundance merely because of the immunologic/allergic background of the patient, presumably representing an unusual cellular response.

In summary, the term eosinophilic cholecystitis has been used highly variably and perhaps too liberally in the literature. Our study, based on a detailed microscopic examination of a large number of consecutive cases, makes a more precise determination of the frequency of eosinophilic cholecystitis and its associations. We recommend reserving this term for cases showing massive eosinophilic infiltrates to a degree of exclusion or dwarfing of other inflammatory cells and also involving multiple layers of the gallbladder. If defined as such, eosinophilic cholecystitis is a very uncommon condition, accounting for only 0.95% of cholecystectomies. Such cases may also have an allergic basis. Accordingly, our findings indicate the likelihood of two potentially distinct mechanisms for the participation of eosinophils in gallbladder injury. First, as a local response, a form of chemical cholecystitis occurs in the setting of acute and subacute disease associated with breached epithelial integrity leading to exposure of the gallbladder wall to the effects of the bile and may be a trigger for the chemotaxis of eosinophils. Second, in the setting of a systemic tendency to generate an eosinophilic response, typically in younger women, having heavy transmural eosinophilic infiltrates in relatively uninjured gallbladders, usually in the paucity of other inflammatory cells. These cases, we believe, qualify to be designated as eosinophilic cholecystitis. 

## Figures and Tables

**Figure 1 diagnostics-13-02559-f001:**
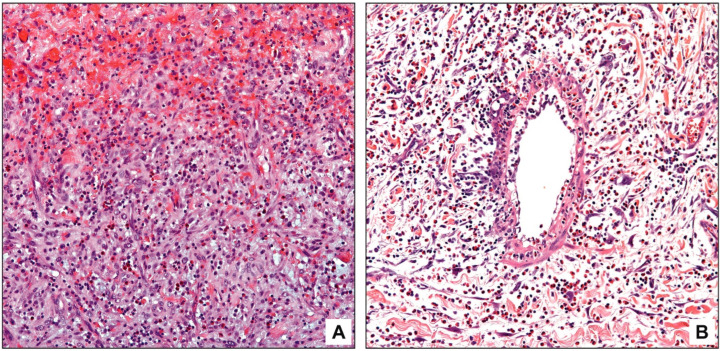
Eosinophils in acute cholecystitis. Eosinophils are often in the hemorrhagic foci (**A**). In some cases, secondary vasculitis by eosinophils is also observed (**B**).

**Figure 2 diagnostics-13-02559-f002:**
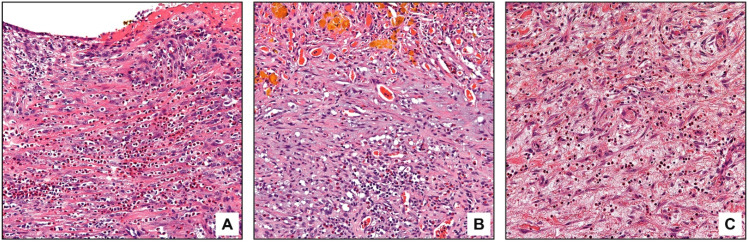
Eosinophils in subacute cholecystitis. Eosinophils are often concentrated in areas of ulceration (**A**) or impacted stones (**B**). Foci of subacute injury are characterized by edema and young tissue culture-like fibroblasts; they also have noticeable eosinophils (**C**).

**Figure 3 diagnostics-13-02559-f003:**
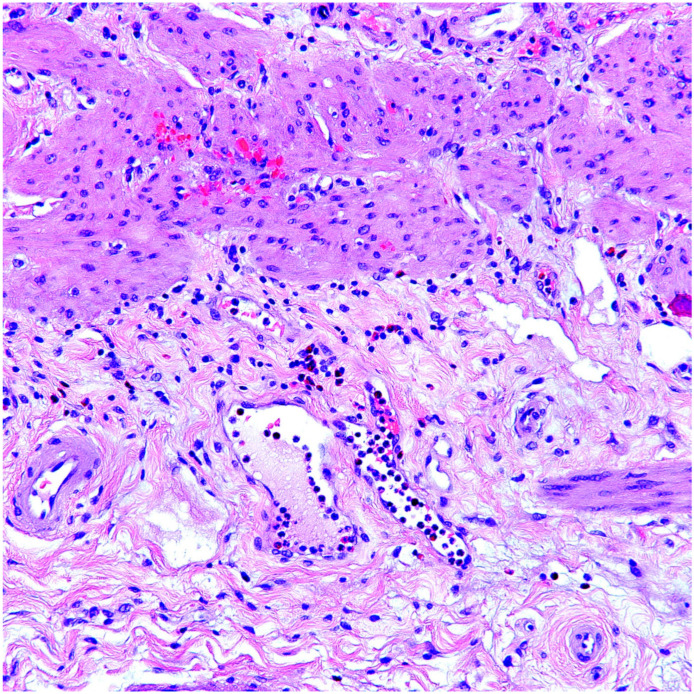
Eosinophils in chronic cholecystitis. Eosinophils are usually very sparse and localized to the deep edges of the muscularis.

**Figure 4 diagnostics-13-02559-f004:**
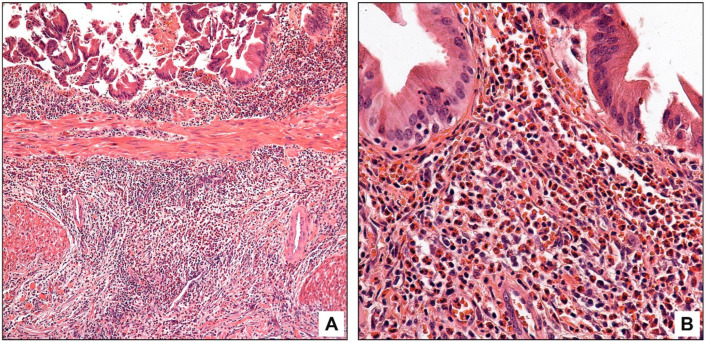
Eosinophilic cholecystitis with a focally enhanced (patchy) pattern characterized by multiple scattered hot spots that have >100 eosinophils/HPF; low magnification (**A**), high magnification (**B**).

**Figure 5 diagnostics-13-02559-f005:**
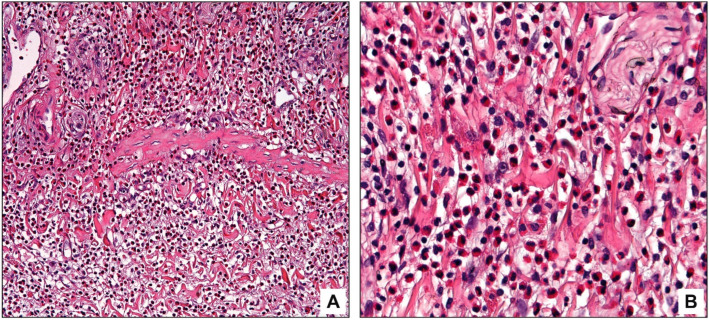
Eosinophilic cholecystitis with a diffuse pattern characterized by intense eosinophil infiltration, forming sheets, to the exclusion of other inflammatory cells; low magnification (**A**), high magnification (**B**).

**Table 1 diagnostics-13-02559-t001:** Comparison of chronic, subacute, and acute cholecystitis cases in terms of clinicopathologic characteristics.

	Acute Cholecystitis (*n* = 55)	Subacute Cholecystitis (*n* = 100)	Chronic Cholecystitis (*n* = 895)		*p* Value
			Ordinary Chronic Cholecystitis (*n* = 885)	Eosinophilic Cholecystitis (*n* = 10)	
Mean age (year)	52.5	51	49	43	0.23
Female/Male	1.5	2.5	3	9	0.05
Cholelithiasis (%)	72	79	82	Patchy pattern: 100 Diffuse pattern: 50	0.85
Mean wall thickness of cases with noticeable eosinophils (mm)	6.6	5.9	4.2	Patchy pattern: 3.5 Diffuse pattern: 3.2	<0.001

## Data Availability

The datasets generated during and/or analyzed during the current study are not publicly available but are available from the corresponding author upon reasonable request.

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
