# Peer review of "Eosinophilic Cholecystitis and Eosinophils in Gallbladder Injuries: A Clinicopathological Analysis of 1050 Cholecystectomies"

_diagnostics, 2023, doi:10.3390/diagnostics13152559_

Round 1
Reviewer 1 Report
Dear Memis and colleagues,
Thank you. I had the pleasure in reading your manuscript; "EOSINOPHILIC CHOLECYSTITIS” AND EOSINOPHILS IN 2 GALLBLADDER INJURY: A CLINICOPATHOLOGICAL 3 ANALYSIS OF 1050 CHOLECYSTECTOMIES".
I have a comment,
Did any of your patients have systemic eosinophilia or related drug injury? I realise that you have mentioned that none had eosinophlia syndromes.
Best wishes
Author Response
None of our patients had known systemic eosinophilia. Six patients had drug reactions. This information is provided in Results section (Page 3, Lines 121-123).
Reviewer 2 Report
INTRODUCTION
- First paragraph: it provides very general concepts ,and the authors use 16 references, which are definitely too many. I think 4-5 references are enough for such a general introduction. Moreover, many are not very recent.
- Moreover, I do not think this first paragraph is very explanatory and/or informative. The authors should provide clear definition of GI eosinophilic disorders and a brief, but clear, overview of this pathological setting, in order to give preliminary concepts for the core topic of the study (namely, gallbladder involvement and its characteristics in this pathological setting).
- Notably, following the previous comments, in the introduction the authors use a total of 31 references out of a total of 39 for the whole manuscript, which is unusual.
- Overall, I think that the introduction should be completely re-organized and should be more pertinent and focused on the core topic of the study.
METHODS
- the ethical statement is insufficient. IRB name(s), IRB approval number(s), and IRB approval date(s) should be provided in this subsection. Moreover, there is no mention about the procedure of informed consent (or its waiver by the IRB).
- Actually, this information is partially provided in one section before the references. However, in the text, it is not specified the hospital site where study participants were recruited.
- Anyway, looking at the authorship (with names from 7 different institutions all around the world), is not really clear where the study was performed. Assuming this is a mono-centric study (based on the only IRB approval stated by the authors), it is not clear the individual authors’ contribution, which is not stated at all.
- The study population is represented by the patients from whom these specimens are coming from. However, the authors do not provide clear inclusion and exclusion criteria.
- Notably, the study period is not provided at all. Therefore, it is not even clarified if all consecutive specimens were retrieved.
- As regards acute/subacute cholecystitis is not clear if were both calculous and acalculous. Anyway, it is unclear the definition of acute and subacute from the clinical point of view.
- The description of statistical analysis is not complete.
- Which histopathological staining were available? Just the original slides were retrieved or some fixed specimens were re-used and stained?
- Therefore, this section needs important specification and should be expanded.
After these preliminary manuscript rearrangements and methodological clarification, I am available for additional cycle(s) of review.
INTRODUCTION
- First paragraph: it provides very general concepts ,and the authors use 16 references, which are definitely too many. I think 4-5 references are enough for such a general introduction. Moreover, many are not very recent.
- Moreover, I do not think this first paragraph is very explanatory and/or informative. The authors should provide clear definition of GI eosinophilic disorders and a brief, but clear, overview of this pathological setting, in order to give preliminary concepts for the core topic of the study (namely, gallbladder involvement and its characteristics in this pathological setting).
- Notably, following the previous comments, in the introduction the authors use a total of 31 references out of a total of 39 for the whole manuscript, which is unusual.
- Overall, I think that the introduction should be completely re-organized and should be more pertinent and focused on the core topic of the study.
METHODS
- the ethical statement is insufficient. IRB name(s), IRB approval number(s), and IRB approval date(s) should be provided in this subsection. Moreover, there is no mention about the procedure of informed consent (or its waiver by the IRB).
- Actually, this information is partially provided in one section before the references. However, in the text, it is not specified the hospital site where study participants were recruited.
- Anyway, looking at the authorship (with names from 7 different institutions all around the world), is not really clear where the study was performed. Assuming this is a mono-centric study (based on the only IRB approval stated by the authors), it is not clear the individual authors’ contribution, which is not stated at all.
- The study population is represented by the patients from whom these specimens are coming from. However, the authors do not provide clear inclusion and exclusion criteria.
- Notably, the study period is not provided at all. Therefore, it is not even clarified if all consecutive specimens were retrieved.
- As regards acute/subacute cholecystitis is not clear if were both calculous and acalculous. Anyway, it is unclear the definition of acute and subacute from the clinical point of view.
- The description of statistical analysis is not complete.
- Which histopathological staining were available? Just the original slides were retrieved or some fixed specimens were re-used and stained?
- Therefore, this section needs important specification and should be expanded.
After these preliminary manuscript rearrangements and methodological clarification, I am available for additional cycle(s) of review.
Author Response
INTRODUCTION
- First paragraph: it provides very general concepts, and the authors use 16 references, which are definitely too many. I think 4-5 references are enough for such a general introduction. Moreover, many are not very recent.
As per the reviewer’s recommendation, original references #1-16, 20, 21, and 23 removed from the Introduction.
- Moreover, I do not think this first paragraph is very explanatory and/or informative. The authors should provide clear definition of GI eosinophilic disorders and a brief, but clear, overview of this pathological setting, in order to give preliminary concepts for the core topic of the study (namely, gallbladder involvement and its characteristics in this pathological setting).
We thank the reviewer for this insightful comment and agree that the first paragraph of the introduction does not add much to the manuscript. Therefore, we eliminated the first paragraph and modified the reminder of the introduction.
- Notably, following the previous comments, in the introduction the authors use a total of 31 references out of a total of 39 for the whole manuscript, which is unusual.
Since there have been different definitions/criteria to define GI eosinophilic disorders in the literature; originally, we had tried to cite all the relevant manuscripts. However, as per the reviewer’s recommendation, we removed the relatively old ones (original references #1-16, 20, 21, and 23) from the Introduction.
METHODS
- The ethical statement is insufficient. IRB name(s), IRB approval number(s), and IRB approval date(s) should be provided in this subsection. Moreover, there is no mention about the procedure of informed consent (or its waiver by the IRB).
As per the reviewer’s recommendation, the ethical statement is updated (Page 2, Lines 65-68 and Page 8, Lines 270-271).
“The study was conducted in accordance with the Declaration of Helsinki and approved by the Institutional Review Board of Emory University (IRB name - Clinicopathologic and Molecular Analysis of Gallbladder and Biliary Tract Lesions; IRB approval number - IRB00010713; protocol code - 8/1/2008 and date of approval - 23 May 2017).”
The procedure of informed consent form was provided to the journal.
- Actually, this information is partially provided in one section before the references. However, in the text, it is not specified the hospital site where study participants were recruited.
Study participants were recruited at Emory University Hospital. This information is added to the text (Page2, Line 70).
“Original H&E stained histologic sections of 1050 consecutive benign cholecystitis cases accessioned at Emory University Hospital …”
- Anyway, looking at the authorship (with names from 7 different institutions all around the world), is not really clear where the study was performed. Assuming this is a mono-centric study (based on the only IRB approval stated by the authors), it is not clear the individual authors’ contribution, which is not stated at all.
Since it was not requested by the journal, we did not document each author’s contribution to the study in the manuscript. Please see below paragraph explaining each author’s contribution.
The study was performed at Emory University and led by Dr. Adsay. Some of authors are either Dr. Adsay’s former colleagues (Michelle Reid) or former international research associates (Bahar Memis, Burcu Saka, Pelin Bagci) who worked at Emory University while the study was being performed. Others are visiting physicians (Juan Carlos Roa, Sudeshna Bandyopadhay, Olca Basturk) or current colleagues (Berk Kaan Aktas, Ayse Armutlu) who participated to the study at different stages.
Conceptualization: Volkan Adsay, Olca Basturk.
Data curation: Bahar Memis, Burcu Saka, Pelin Bagci, Juan Carlos Roa
Histopathologic Review: Volkan Adsay, Bahar Memis, Burcu Saka, Pelin Bagci, Michelle Reid
Literature search/investigation: Sudeshna Bandyopadhay, Berk Kaan Aktas, Ayse Armutlu
Writing–originaldraft: Bahar Memis, Berk Kaan Aktas, Ayse Armutlu
Writing–review&editing: Volkan Adsay, Olca Basturk, Pelin Bagci, Michelle Reid, Juan Carlos Roa, Sudeshna Bandyopadhay
Project administration: Volkan Adsay, Olca Basturk
- The study population is represented by the patients from whom these specimens are coming from. However, the authors do not provide clear inclusion and exclusion criteria.
As per the reviewer’s recommendation, this information is added to the text (Page 2, Lines 70-76).
“… consecutive benign cholecystitis cases accessioned at Emory University in 2002 to 2010 were retrieved and reevaluated …”.
“Cholecystectomies performed during bypass/banding operations or pancreatoduodenectomies or cholecystectomies with neoplastic lesions such as dysplasia, intracholecystic papillary neoplasm or invasive carcinoma were excluded.”
- Notably, the study period is not provided at all. Therefore, it is not even clarified if all consecutive specimens were retrieved.
As per the reviewer’s recommendation, the study period is added to the text (Page 2, lines 70, 71).
“Original H&E stained histologic sections of 1050 consecutive benign cholecystitis cases accessioned at Emory University in 2002 to 2010 were retrieved and reevaluated to determine …”
- As regards acute/subacute cholecystitis is not clear if were both calculous and acalculous. Anyway, it is unclear the definition of acute and subacute from the clinical point of view.
72% of the acute cholecystitis and 79% of the subacute cholecystitis cases had cholelithiasis. This information is provided in Table 1.
Acute and subacute cholecystitis terms were used in the text to describe morphologic changes, not clinical presentation.
- The description of statistical analysis is not complete.
As per the reviewer’s recommendation, the description of statistical analysis is updated in the text (Page 3, lines 102-106).
- Which histopathological staining were available? Just the original slides were retrieved or some fixed specimens were re-used and stained?
Original H&E stained histologic sections of 1050 consecutive benign cholecystitis cases were retrieved and reevaluated for the study. This information was added to the text (Page 2, lines 70, 71).
Reviewer 3 Report
Nice scientific paper on o subject non very studied before. The paper respects the usual structural frame with nice detailed results. In the future, I would try to make correlations between eosinophilic cholecystitis and eosinophilic esophagitis and colitis
Author Response
We thank the reviewer for the the positive comments on our paper.
Round 2
Reviewer 2 Report
Unfortunately, the authors have not appropriately or correctly addressed most of my previous comments. See RR
INTRODUCTION
R- First paragraph: it provides very general concepts, and the authors use 16 references, which are definitely too many. I think 4-5 references are enough for such a general introduction. Moreover, many are not very recent.
A- As per the reviewer’s recommendation, original references #1-16, 20, 21, and 23 removed from the Introduction.
RR-First of all, let me highlight that a revised manuscript in word-review mode as it is usual practice for the journal would have been much more appropriate. As regards, the number of references, the authors reduced the number, even if these seems still too many, considering that the introduction is quite short. By the way, the changes in references are not tracked. Moreover, I think the introduction should be more informative about the definition and classification (both clinical and pathological) of cholecystitis.
R - Moreover, I do not think this first paragraph is very explanatory and/or informative. The authors should provide clear definition of GI eosinophilic disorders and a brief, but clear, overview of this pathological setting, in order to give preliminary concepts for the core topic of the study (namely, gallbladder involvement and its characteristics in this pathological setting).
A- We thank the reviewer for this insightful comment and agree that the first paragraph of the introduction does not add much to the manuscript. Therefore, we eliminated the first paragraph and modified the reminder of the introduction.
RR- As above, we cannot see any tracked correction in the introduction and it does not seem that the authors have made any significant improvement.
R - Notably, following the previous comments, in the introduction the authors use a total of 31 references out of a total of 39 for the whole manuscript, which is unusual.
A- Since there have been different definitions/criteria to define GI eosinophilic disorders in the literature; originally, we had tried to cite all the relevant manuscripts. However, as per the reviewer’s recommendation, we removed the relatively old ones (original references #1-16, 20, 21, and 23) from the Introduction.
RR- Unfortunately, the changes can be completely assessed due to the absence of tracked corrections. However, the introduction is quite poor.
METHODS
R - The ethical statement is insufficient. IRB name(s), IRB approval number(s), and IRB approval date(s) should be provided in this subsection. Moreover, there is no mention about the procedure of informed consent (or its waiver by the IRB).
A- As per the reviewer’s recommendation, the ethical statement is updated (Page 2, Lines 65-68 and Page 8, Lines 270-271).
“The study was conducted in accordance with the Declaration of Helsinki and approved by the Institutional Review Board of Emory University (IRB name - Clinicopathologic and Molecular Analysis of Gallbladder and Biliary Tract Lesions; IRB approval number - IRB00010713; protocol code - 8/1/2008 and date of approval - 23 May 2017).”
The procedure of informed consent form was provided to the journal.
RR- The authors should specify all the clinical sites involved in the data collection and then the related IRB approval. The procedure and type of informed consent must be stated and explained in the manuscript for complete transparency.
R - Actually, this information is partially provided in one section before the references. However, in the text, it is not specified the hospital site where study participants were recruited.
A- Study participants were recruited at Emory University Hospital. This information is added to the text (Page2, Line 70).
“Original H&E stained histologic sections of 1050 consecutive benign cholecystitis cases accessioned at Emory University Hospital …”
RR- If only one hospital site was involved in the patients’ (data) recruitment, can you explain why you reported 3 ethical approvals?
R - Anyway, looking at the authorship (with names from 7 different institutions all around the world), is not really clear where the study was performed. Assuming this is a mono-centric study (based on the only IRB approval stated by the authors), it is not clear the individual authors’ contribution, which is not stated at all.
A - Since it was not requested by the journal, we did not document each author’s contribution to the study in the manuscript. Please see below paragraph explaining each author’s contribution.
The study was performed at Emory University and led by Dr. Adsay. Some of authors are either Dr. Adsay’s former colleagues (Michelle Reid) or former international research associates (Bahar Memis, Burcu Saka, Pelin Bagci) who worked at Emory University while the study was being performed. Others are visiting physicians (Juan Carlos Roa, Sudeshna Bandyopadhay, Olca Basturk) or current colleagues (Berk Kaan Aktas, Ayse Armutlu) who participated to the study at different stages.
Conceptualization: Volkan Adsay, Olca Basturk.
Data curation: Bahar Memis, Burcu Saka, Pelin Bagci, Juan Carlos Roa
Histopathologic Review: Volkan Adsay, Bahar Memis, Burcu Saka, Pelin Bagci, Michelle Reid
Literature search/investigation: Sudeshna Bandyopadhay, Berk Kaan Aktas, Ayse Armutlu
Writing–originaldraft: Bahar Memis, Berk Kaan Aktas, Ayse Armutlu
Writing–review&editing: Volkan Adsay, Olca Basturk, Pelin Bagci, Michelle Reid, Juan Carlos Roa, Sudeshna Bandyopadhay
Project administration: Volkan Adsay, Olca Basturk
RR- Thank you. I cannot fully judge the contributions, so I will the editor to judge. However, this information is required by MDPI journal to be published at the end of the manuscript.
R - The study population is represented by the patients from whom these specimens are coming. However, the authors do not provide clear inclusion and exclusion criteria.
A- As per the reviewer’s recommendation, this information is added to the text (Page 2, Lines 70-76).
“… consecutive benign cholecystitis cases accessioned at Emory University in 2002 to 2010 were retrieved and reevaluated …”.
“Cholecystectomies performed during bypass/banding operations or pancreatoduodenectomies or cholecystectomies with neoplastic lesions such as dysplasia, intracholecystic papillary neoplasm or invasive carcinoma were excluded.”
RR- Please, can you explain the expression “BENIGN” cholecystitis, since this is not clear to me.
R- Notably, the study period is not provided at all. Therefore, it is not even clarified if all consecutive specimens were retrieved.
A- As per the reviewer’s recommendation, the study period is added to the text (Page 2, lines 70, 71).
“Original H&E stained histologic sections of 1050 consecutive benign cholecystitis cases accessioned at Emory University in 2002 to 2010 were retrieved and reevaluated to determine …”
RR- Accepted.
R - As regards acute/subacute cholecystitis is not clear if were both calculous and acalculous. Anyway, it is unclear the definition of acute and subacute from the clinical point of view.
A- 72% of the acute cholecystitis and 79% of the subacute cholecystitis cases had cholelithiasis. This information is provided in Table 1.
Acute and subacute cholecystitis terms were used in the text to describe morphologic changes, not clinical presentation.
RR- Then, all this information and explanations should be clarified in the methods and introduction. The classificatory terms should precisely defined.
R - The description of statistical analysis is not complete.
A- As per the reviewer’s recommendation, the description of statistical analysis is updated in the text (Page 3, lines 102-106).
RR- please, can you clarify where you used ANOVA? Then, why did you choose that specific post-hoc test? Did you check the normality of the distributions??
R - Which histopathological staining were available? Just the original slides were retrieved or some fixed specimens were re-used and stained?
A- Original H&E stained histologic sections of 1050 consecutive benign cholecystitis cases were retrieved and reevaluated for the study. This information was added to the text (Page 2, lines 70, 71).
RR- accepted.
Improvement is needed.
Author Response
Dear Editor, as per your request, we provided more detailed methodology and answered your questions.
I also want to let you know that we are more than happy to expand the main text more to have a minimum word count of 4000 words. However, unfortunately Reviewer #2 still seem to have some major issues with our paper even though Reviewers #1 and #3 requested very minor revisions and made positive comments on our paper.
During Round 1, we tried to do our best to answer his/her questions and made all the changes he/she requested despite his/her unprofessional and offensive tone. To our disappointment, the reviewer is getting increasingly prejudicial and unacceptable and now
1) criticizing us for following the journal’s instructions such as using the formatted version of the manuscript to make the revisions requested
2) demanding explanations for things that did not even happen such as asking why we reported 3 ethical approvals even though we submitted only 1
3) seems to be obsessed with the # of the references we used in our introduction (we simply wanted to include all the relevant references in the literature as the definitions and findings have been controversial in the literature. Even though we reduced the # of the references significantly he/she wants us to do eliminate more. However, if we do what the reviewer wants (include only 4-5 references) we would not be able to demonstrate the controversies in the literature.
4) complaining about our introduction being short this time but it was the reviewer’s request to delete certain parts of the introduction in the first place
5) implying our authorship decisions are inappropriate (“I cannot fully judge the contributions, so I will the editor to judge. However, this information is required by MDPI journal to be published at the end of the manuscript”. As you know, we had provided this information to the journal during the original submission but made more detailed explanation again because the reviewer asked for it one more time and now, he/she sounds like he/she is not happy about how many authors we have. Please keep in mind that this study has a very large cohort, and it really took a team to finish all the work).
6) for someone whose both reviews are full of grammatical mistakes, typos and sentences that are wrong (such as “Unfortunately, the changes can be completely assessed due to the absence of tracked corrections”), we believe he/she is not in a position to criticize the English of our manuscript.
I can go on but I believe I made my point.
Therefore, I believe that this reviewer has some personal issues, not to mention is unprofessional, and wanted to check with you first to see if you are willing to ignore this reviewer’s second round requests as I do not want to subject my co-authors (a mixed group of pathologists from multiple countries) to the same inappropriate and biased review one more time. I also do not wish to waste either our valuable time or yours for that matter.
If this request is not acceptable for you, please let us know and we will withdraw our manuscript and submit it to another journal.
Sincerely,
Olca Basturk